# New In Vitro Methodology for Kinetics Distribution Prediction in the Brain. An Additional Step towards an Animal-Free Approach

**DOI:** 10.3390/ani11123521

**Published:** 2021-12-10

**Authors:** Bárbara Sánchez-Dengra, Isabel González-Álvarez, Marta González-Álvarez, Marival Bermejo

**Affiliations:** Pharmacokinetics and Pharmaceutical Technology Area, Department of Engineering, Miguel Hernandez University, 03550 Alicante, Spain; barbarasanchezdengra@gmail.com (B.S.-D.); isabel.gonzalez@umh.es (I.G.-Á.); mbermejo@umh.es (M.B.)

**Keywords:** blood–brain barrier (BBB), unbound fraction (f_u_), distribution volume in brain (V_u,brain_), 3Rs

## Abstract

**Simple Summary:**

The prevalence of neurological disorders in humans is rising year after year. This fact necessitates the development of new drugs for treating these pathologies. Traditionally, drugs have been tested in animals prior to use in human experiments; however, the use of animals in experimentation must be controlled and as low as possible. Because of that, here we proposed a new in vitro approach with which the access and distribution of drugs into the brain can be evaluated without using/killing any animals.

**Abstract:**

The development of new drugs or formulations for central nervous system (CNS) diseases is a complex pharmacologic and pharmacokinetic process; it is important to evaluate their access to the CNS through the blood–brain barrier (BBB) and their distribution once they have acceded to the brain. The gold standard tool for obtaining this information is the animal microdialysis technique; however, according to 3Rs principles, it would be better to have an “animal-free” alternative technique. Because of that, the purpose of this work was to develop a new formulation to substitute the brain homogenate in the in vitro tests used for the prediction of a drug’s distribution in the brain. Fresh eggs have been used to prepare an emulsion with the same proportion in proteins and lipids as a human brain; this emulsion has proved to be able to predict both the unbound fraction of drug in the brain (f_u,brain_) and the apparent volume of distribution in the brain (V_u,brain_) when tested in in vitro permeability tests. The new formulation could be used as a screening tool; only the drugs with a proper in vitro distribution would pass to microdialysis studies, contributing to the refinement, reduction and replacement of animals in research.

## 1. Introduction

Neurological disorders are getting more and more frequent due to global aging. In fact, it is estimated that in 2050, 22% of people worldwide will be over 60 years old [1]. After that age, several physiological processes, such as, lower levels of acetylcholine, dopaminergic and cholinergic neurons, the accumulated DNA mutations and the presence of other comorbidities, like, obesity, diabetes, hypertension or hyperlipidaemia, can lead to an increase of neurodegenerative disorders (dementia, Alzheimer’s disease or Parkinson’s disease), brain tumors (glioblastoma), cerebral stroke, epilepsy or depression [2]. Table 1 shows the global levels of prevalence of neurological disorders in 2000 and 2019 for people of all ages and people from 60 to 89 years old (this data was obtained from the GBD online results tool [3]).

According to Table 1, in general, in the last decade, the prevalence of all neurological disorders has increased by at least 30%; however, this increment gets much more pronounced in the population over 60 years old, with a minimum increment of 65% [3]. One could think that the increments in prevalence may not be significant, because the total population in the world has also increased with time, moving from 6143.5 million people in 2000 to 7713.5 million people in 2019 for all ages, and from 602.7 to 996.7 million people for the group from 60 to 89 years. Nonetheless, as can be seen in the column Norm_∆, where prevalence is normalized, there is a considerable increase in almost all neurological disorders.

Treatment of brain diseases requires drugs that are able to reach brain targets; because of that, an extremely high number of molecules and formulations need to be studied to get a successful treatment for neurological disorders [4]. The common failures in the development of CNS drugs are lack of activity or, more commonly, lack of biopharmaceutical suitable properties. Furthermore, the current screening methods for blood–brain barrier accessibility lack high-throughput capacity and rely on the intensive use of animal models or tissues.

When a new treatment that needs to reach the central nervous system (CNS) is developed, the gold standard for measuring the concentration it reaches in the different parts of the brain is microdialysis. Microdialysis allows researchers to measure the unbound concentration of drug at different times in a specific brain area and, although this measurement can be done in humans, it is more common to measure the levels in rats or mice and then translate the information into human brain levels using physiologically based pharmacokinetic (PBPK) modeling [5,6].

In brief, when the microdialysis technique is used for measuring drug levels in the CNS, a small cannula with an inner and an external conduct is introduced in the animal’s brain. Then, a saline solution (perfusate) is supplied through the internal conduct and, when it gets in touch with the brain in the external conduct, which has a semipermeable membrane, it starts to mix with the components present in the extracellular fluid (ECF) of the CNS, because substances with a diameter smaller than the membrane pores, diffuse from the more concentrated solution to the less concentrated one. Finally, the “mixed” solution (dialysate) is recovered through the same cannula and, at different times, the amount of drug present in it is analyzed [7]. In Figure 1, a scheme of this system is shown.

The unbound concentration profiles obtained by microdialysis give, among others, information about the unbound fraction of drug in the brain (f_u,brain_) and the apparent volume of distribution in the brain (V_u,brain_). Both parameters are interrelated and are considered crucial when studying a new drug for the treatment of a neurological disorder. On the one hand, f_u,brain_ is an important parameter, as only the free drug is able to cross membranes, so the free fraction of drug is the one that will contribute to the equilibrium between blood and brain through the blood–brain barrier (BBB) and, furthermore, it will be the only one able to enter into the cells or to bind its target [8]. On the other hand, V_u,brain_ reflects the drug binding to the brain with independence of the BBB equilibrium; the value of this parameter can be compared with the physiological volumes of the CNS to study the drug affinity to the brain tissue [8]. If a drug is highly permeable through the BBB and, at the same time, has a high V_u,brain_, it is more probable that it will perform properly.

Due to the high number of molecules and formulations to test, a great number of animals are required for these assays, along with the attendant ethical problems of experimenting on animals. Moreover, there are some issues with the translation of data from animals to humans; however, the use of human biology-based in vitro methods is vital for better understanding human diseases. Thus, the objective of the present work is to propose an innovative in vitro method amenable to high-throughput testing and which substitutes the use of brain animal/human homogenate in accordance to the 3 R’s principles (replacement, reduction and refinement), which were established in the 20th century by Russell and Burch and which, nowadays, are strictly followed by scientists all over the world [9]. In fact, nowadays, several legislative documents that regulate animal experimentation can be found in the majority of animal testing countries.

Having in mind the 3Rs principles, several in vitro and in silico methods have been proposed to study the access and distribution of drugs into the brain. In vitro methods can be classified according to its base in: non-cell based in vitro methods, such as, PAMPA-BLM (black lipid membrane) or PAMPA-BBB models [10,11], and cell based in vitro methods, like, MDCK, MDCK-MDR1, Caco-2 or hCMEC/D3 cell lines [12,13]. All these in vitro methods have demonstrated that they are capable of predicting the permeability clearance into the brain (Cl_in_) of most of the molecules (the passive ones). For instance, in 2010, the PAMPA-BBB model was able to classify 13 compounds out of 14 in BBB+ or BBB- according to their in vivo LogBB (logarithm of the ratio of the steady-state total concentration of a compound in the brain to that in the plasma), and the misclassified compound was corrected using Caco-2 monolayers [14], as the Cl_in_ also depends on the transporters present on the BBB (not present in PAMPA models).

Nonetheless, knowing the Cl_in_ of a drug on its own is not enough, as only the unbound fraction of drug in the brain will be able to reach its target and give an effect. Taking two opioid drugs, morphine and loperamide, as an example, and consulting the bibliography for its ability to access CNS, it can be seen that morphine (10.4 ± 3 μL/min × g brain) has a lower Cl_in_ than loperamide (98.6 ± 17.3 μL/min × g brain), so morphine has a lower permeability through the BBB [15,16]. Despite that, morphine is a much more potent drug, so Cl_in_ alone is not a good metric for potency.

Cases such as the one mentioned above show that in order to correctly determine whether a drug will carry out its function in the CNS, it is necessary, in addition to its access, to know its distribution once it has crossed the BBB. This distribution is defined by the previously mentioned parameters (f_u,brain_ and V_u,brain_). Thus, in 2013, Mangas-Sanjuan et al. [17] designed a new in vitro system with which the f_u,brain_ the V_u,brain_, and also the unbound plasma-brain partition coefficient (Kp_uu,brain_) and the unbound fraction of drug in plasma (f_u,plasma_), could be obtained. A scheme of this system which, some years later, was tested and validated using another cell line [18], is shown in Figure 2.

In this in vitro approach, the f_u,brain_ and V_u,brain_ parameters are obtained by means of the combination of the permeability values from two basolateral-to-apical experiments, a standard one and one modified with brain homogenate (Figure 2) [17,18]. The system meets the reduction principle of the 3Rs, as the brain homogenate of the same animal can be divided and used in several wells, a fact that can be considered a great advance in the techniques used in the development of drugs for the CNS, as an extremely high number of molecules and formulations need to be studied to get a successful treatment for neurological disorders [4].

However, although the model proposed in 2013 by Mangas-Sanjuan et al. [17] could be used as a screening tool that would reduce the number of animals used in neurological research, it still uses brain homogenate from pigs. Thereby, with the aim of improving the model and meeting the replacement principle of the 3Rs, the purpose of this work was to develop a new formulation, based on unfertilized chicken eggs, to substitute the brain homogenate of the system and create an “animal-free” in vitro screening tool, able to predict both the f_u,brain_ and V_u,brain_ parameters.

## 2. Materials and Methods

### 2.1. Drugs and Products

The nine drugs tested (amitryptiline, atenolol, carbamazepine, fleroxacin, loperamide, norfloxacin, pefloxacin, propranolol and zolpidem) and HPLC grade solvents (acetonitrile, methanol and water) were purchased from Sigma-Aldrich (Barcelona, Spain). MDCK cell line was purchased from ATCC (USA) and MDCK-MDR1 cells were provided by Dr. Gottessman, MM (Nathional Institutes of Health, Bethesda). Pig brain homogenate was kindly supplied by a local slaughterhouse and fresh unfertilized chicken eggs were bought in a local supermarket. Table 2 shows the molecular properties of the nine drugs mentioned above.

Dulbecco’s modified Eagle’s medium (DMEM) with high content of glucose, L-glutamine, HEPES, MEM non-essential aminoacid, penicillin−streptomycin, trypsin-EDTA, Hank’s balanced salt solution (HBSS) and fetal bovine serum (FBS) for the cell culture of MDCK and MDCK-MDR1 cell lines were purchased from Sigma-Aldrich.

### 2.2. Preparation of the Brain Homogenate and the New Formulation for Substituting It

The brain homogenate was obtained after triturating the pig brains that were kindly supplied by a local slaughterhouse and mixing them with phosphate buffer (180 mM, pH 7.4) solution in a ratio 1:3 (brain:buffer).

For preparing 100 g of the new formulation for substituting the brain homogenate, 2 medium size eggs, whose weight without the shell is around 100 g [21], were crushed; the whites and the yolks were separated. Then, 15.35 g of whites were mixed with 67.73 mL of water. In another beaker, 16.92 g of yolk were weighed separately. The yolk was poured into the white-water mixture and stirred vigorously until obtaining an emulsion.

### 2.3. Cell Culture and Permeability Studies

The permeability studies were carried out in two different cell lines: MDCK and MDCK-MDR1. MDCK and MDCK-MDR1 cells come from the kidney of dogs; however, when they are properly cultured, they form monolayers with quite strong and tight junctions [22,23]. It is for that reason that they are accepted as appropriate tools to simulate the BBB, although they do not have BBB transporters. In the MDCK-MDR1 cell line, the issue of the lack of transporters is partially solved with the transfection with P-glycoprotein (Pgp), the most common efflux transporter in the BBB; thus, this line would be ideal for studying drugs with a passive access to the CNS, as well as drugs which are substrates of Pgp, while MDCK would be better for studying passive drugs.

Both types of cells were cultured and seeded following the protocol explained in [17]. When the monolayers were confluent, three types of experiments, from the basolateral to the apical chamber, were carried out:Standard BA: In this experiment, drugs previously dissolved in HBSS at the concentration shown in Table 3, were placed at the basolateral chamber. After taking the samples and making the necessary calculations, the apparent efflux permeability (P_app B→A_) was obtained from this experiment.Brain homogenate BA: In this case, the free drug apparent efflux permeability (P_app HOM_) was obtained after adding the drug dissolved in a 1:3 pig brain homogenate:phosphate buffer (180 mM, pH 7.4) solution to the basolateral chamber.Emulsion BA: Finally, in this third condition, as it is the equivalent to the brain homogenate BA experiment, but using the new formulation as a substitute of brain homogenate, the parameter obtained was also the free drug apparent efflux permeability, but in this case labeled as P_app EMUL_.

In all cases, cells were seeded in 6-transwell plates (effective area: 4.2 cm^2^, pore size: 0.4 micron and pore density: (100 ± 10) × 10^6^/cm^2^) and its transepithelial electrical resistance (TEER) was measured before and after the experiments, considering that cells were confluent when the TEER values reach around 130–150 kΩ·cm^2^ for MDCK cells and around 120–140 kΩ·cm^2^ for MDCK-MDR1 cells [22]. The apical side of the system was filled in with HBSS and samples were taken at 15, 30, 60 and 90 min. Additionally, for evaluating the mass balance, two samples were taken from basolateral at time 0 and 90 min and one sample was taken after disrupting the membrane with methanol. During the experiments, cells were maintained in an orbital shaker at 37 °C and 100 rpm, so the agitation prevents the drug from precipitating and reduces the formation of a non-stirred layer over the cells, which would decrease the apparent permeability. Once the experiments were finished, samples were frozen at −20 °C, until their analysis.

### 2.4. HPLC Analysis of the Samples

Samples were analyzed by UV-HPLC, using a Waters 2695 separation module, a Waters 2487 UV detector and a column XBridge C18 (3.5 μM, 4.6 × 100 mm); before this, they were centrifuged at 10,000 rpm for 10 min. Additionally, before centrifugation, samples from brain homogenate and emulsion experiments were diluted (50:50) with cold methanol to precipitate proteins. A flow rate of 1 mL/min, a run temperature of 30 °C and an injection volume of 90 μL were defined. The rest of chromatographic conditions that were used are summarized in Table 3. All analytical methods were validated and demonstrated to be adequate regarding linearity, accuracy, precision, selectivity and specificity (see Appendix A for the validation parameters).

### 2.5. Parameters Calculation: P_app_, f_u,brain_ and V_u,brain_

The apparent efflux permeabilities (P_app B→A_, P_app HOM_ and P_app EMUL_) were calculated using the modified non-sink equation [24] (Equation (1)), in which C_r,t_ and C_r,t−1_ are the concentrations in the receiver compartment (in this case, apical) at time t and time t − 1, V_r_ and V_d_ are the volumes of the receiver (apical) and donor (basolateral) compartments, Q_t_ is the total amount of drug in both chambers at time t, f is the sample replacement dilution factor, S is the surface area of the monolayer and Δt is the time interval. P_eff,0_ and P_eff,1_ are the apparent permeability values, which can differ if the permeation rate is different at the beginning of the experiment with regard to the rest of the transport profile.
(1)$$C_{r,\;t}=\frac{Q_{t}}{V_{r}+V_{d}}+\left(\left(C_{r,\;t-1}\cdot f\right)-\frac{Q_{t}}{V_{r}+V_{d}}\right)\cdot e^{-P_{\mathrm{eff}0,1}\cdot S\cdot \left(\frac{1}{V_{r}}+\frac{1}{V_{d}}\right)\cdot \Delta t}$$

P_eff,1_ was the parameter selected to define the apparent efflux permeabilities (P_app B→A_, P_app HOM_ and P_app EMUL_) and with the aid of the Equation (2), they were transformed to f_u,brain_.
(2)$$f_{u,\mathrm{brain}}=\frac{P_{\mathrm{app}\;\mathrm{HOM}}}{P_{\mathrm{app}\;B\to A}}\;\mathrm{or}\;\frac{P_{\mathrm{app}\;\mathrm{EMUL}}}{P_{\mathrm{app}\;B\to A}}$$

Finally, the f_u,brain_ were translated to V_u,brain_ with Equation (3), where V_ECF_ is the volume of extracellular fluid and V_ICF_ is the volume of intracellular fluid. The comparison of the V_u,brain_ with the physiological volumes of the CNS gives an idea of the drug affinity to the brain tissue (the greater the affinity for the tissue, the greater the V_u,brain_) [8].
(3)$$V_{u,\mathrm{brain}}=V_{\mathrm{ECF}}+\frac{1}{f_{u,\mathrm{brain}}}\cdot V_{\mathrm{ICF}}=0.2+\frac{1}{f_{u,\mathrm{brain}}}\cdot 0.6\left(\mathrm{mL}/g\;\mathrm{brain}\right)$$

### 2.6. In Vitro-In Vivo Correlations (IVIVCs): Linear Regression

Both in vitro parameters, f_u,brain_ and V_u,brain_ were related with their correspondent value in vivo to obtain different in vitro–in vivo correlations (IVIVCs). The in vivo data were obtained from the literature, specifically, from the following articles: [25,26]. The IVIVCs were adjusted to a linear model with the following structure: y = a + bx.

## 3. Results

Table 4 shows the apparent permeability values for all the drugs obtained in the different experimental conditions, Table 5 shows the in vitro f_u,brain_ obtained from the different experiments as well as the in vivo values [25,26] for the same parameter. Table 6 is equivalent to Table 5, but for the V_u,brain_ parameter. Results in Table 5 and Table 6 are more visually summarized in Figure 3 and Figure 4.

Figure 3 shows the relationships obtained for the parameter f_u,brain_ when the predicted values obtained with the brain homogenate are compared with the in vivo values for each drug.

Figure 3A,C show the f_u,brain_ predictions for the MDCK cell line when the basolateral chamber is filled in with the brain homogenate or with the new emulsion, respectively, while in Figure 3B,D, the f_u,brain_ predictions with the brain homogenate or with the new emulsion are also shown, but for the MDCK-MDR1 cell line. In the four cases, it can be seen that the smallest values in vitro correspond with the smallest values in vivo and that the biggest values in vitro correspond with the biggest values in vivo.

In Figure 3E,F, in which the predictions from the new emulsion are represented versus the predictions of the brain homogenate, it can be seen that they are more similar when using the MDCK-MDR1 cell line (r^2^ = 0.886).

Figure 4 shows the same relationships as Figure 3, but for the parameter V_u,brain_. In this case, it can be seen that for both cell lines MDCK (4A, 4C and 4E) and MDCK-MDR1 (4B, 4D and 4F), the predictions are quite similar. Moreover, when the new emulsion predictions and the brain homogenate predictions are represented together, the coefficient of determination is higher than 0.900 for both types of cells (r^2^ = 0.978 for MDCK cells and r^2^ = 0.954 for MDCK-MDR1 cells).

## 4. Discussion

Some decades ago, the use of animals in experimentation became controversial; from that moment to nowadays, the search for alternatives to these animals has become necessary [27]. In this work, a new formulation alternative to brain homogenate has been developed as a substitute of this component in the study of the distribution of drugs in the central nervous system.

In an adult human, the CNS weight is around 3% of the total human body weight [28] and, in terms of biochemical composition, the whole human brain is approximately 77–78% water, 10–12% lipids, 8% proteins, 1% carbohydrates, 2% soluble organic substances and 1% inorganic salts [29]. On the other hand, fresh eggs are 12.5% proteins (38% of them in the yolk and 62% in the whites) and 11.1% lipids (all of them in the yolk) [21,30]. Taking into account these concentrations, a new formulation has been prepared in order to obtain an emulsion with the same composition as a human brain; the final concentration of protein has been 8% (5.1% from the yolk + 2.9% from the whites) and the concentration of lipids 12% (from the yolk).

The new formulation has been tested in a previously developed in vitro model [17] with the cell lines MDCK and MDCK-MDR1 (equal to the MDCK cell line, but transfected with P-glycoprotein). Despite their dog kidney origin and their lack of BBB transporters, both cell lines are accepted as appropriate tools to simulate the BBB because, when they are properly cultured, they form monolayers with quite strong and tight junctions [22,23].

Table 2 shows the molecular properties of the drugs used in this study. The selection of the different molecules was done considering their ability to bind P-glycoprotein, their charge at physiological pH, their solubility and their lipophilicity (logP), with the aim of having drugs with different properties. It is because P-glycoprotein is the most relevant transporter in MDCK-MDR1 cells [23] and because the other properties have been previously used in several in silico models (quantitative structure activity relationships–QSAR) when trying to predict the behaviour of drugs in the CNS [31,32,33]. In terms of the concentrations used in the study (Table 3), they were selected according to the ones previously used by Mangas-Sanjuan et al. [17], and the plasma values detected in vivo by Kodaira et al. [26] (when necessary, plasma concentrations were increased until making them detectable by HPLC.)

Figure 3 shows how the new emulsion is able to predict the f_u,brain_ parameter, as well as the brain homogenate, because the coefficients of determination from Figure 3A,B are quite similar to those from Figure 3C,D. A clear tendency can be observed in all the correlations, and the values of r^2^ are consistent to those obtained by Mangas Sanjuan et al. in 2013 (MDCK r^2^ = 0.616 and MDCK-MDR1 r^2^ = 0.624) [17], which although, not published, can be obtained from the V_u,brain_ correlations with Equation (3). The inability to obtain better IVIVCs can be explained by the homogenization process that may denaturalize some proteins of the brain tissue and damage the lipidic structures, altering their binding properties [12]. In a similar way, in the case of the new emulsion, the proportion of lipid and proteins present in brain are respected, but there is not an organized structure in it.

On the other hand, the f_u,brain_ correlations are better for the MDCK-MDR1 cell line, which has an r^2^ of 0.886 when the predictions from brain homogenate and from the new emulsion are represented together (Figure 3F). Probably, this better prediction can be attributed to the presence of the P-glycoprotein in the monolayers of the MDCK-MDR1 cells, as the fraction of drug that binds the efflux transporter does not contribute to the f_u,brain_.

In terms of the V_u,brain_ parameter, predictions from both cell lines are also quite similar as seen in Figure 4. Furthermore, in this case, there is a huge similarity between the predictions of both cell lines (MDCK and MDCK-MDR1), as can be deduced from the higher than 0.900 coefficients of determination from Figure 4E,F. The absence of differences between cell lines can be explained when the definition of the V_u,brain_ parameter is taken into account, since it represents the drug in the brain with independence of the BBB equilibrium [8], thus without being affected by the transporters. As happened with the f_u,brain_ correlations, the low r^2^ in Figure 4A–D, may be explained by the lack of an organized structure in both the brain homogenate and the new emulsion [12].

Despite his success in the prediction of drug brain distribution, this type of in vitro model will not substitute the brain microdialysis technique, at least at the moment, because it cannot reflect all the physiological properties of an alive CNS. Maybe a future approach may be exploring the possibility of developing an organized animal-free slice with an organized structure which could be able to predict f_u,brain_ the V_u,brain_ in a better way. To do this, first, the model should be tested substituting the brain homogenate by brain slices and evaluate its ability to predict brain distribution and then, if it is able to do it properly, the brain slices could be compared with the new animal-free slices. Nevertheless, as it is now, it is a useful tool that can be used in a complementary way when a new drug or an innovative delivery strategy is being developed. Thus, this model can be used as a rapid screening tool and its information, on its own, or combined with other information obtained from in silico [31,32,33] or PBPK [16,34,35] models, could be used to move only a few selected candidates to in vivo studies.

Other in vitro BBB models could also be adapted to use this new “animal-free” (based on unfertilized chicken eggs) emulsion and obtain more information. On the one hand, the cell monolayer could be substituted by a more complex cell line, such as hCMEC/D3, as was previously done by the authors with the Mangas-Sanjuan et al. model [18], for studying new drugs or delivery systems substrate of other transporters different to P-glycoprotein. On the flip side, moving to a simpler way, the new emulsion in combination with PAMPA-BBB or PAMPA-BLM models, which have demonstrated to be able to correctly classify drugs into BBB+ and BBB- groups according to their total concentration [10,11,14], could be used to evaluate not just the access of substances through the BBB, but also their distribution once in the CNS.

It cannot be forgotten that the final aim of this type of study is to try to guess what would happen if the drug is administered to human beings. In this respect, the differences on the specific composition (not in proportion) in lipids and proteins between chickens and humans may hinder the translation of data. Nonetheless, the development of PBPK models has proved to be a helpful tool in this process [16]. Therefore, the data of this paper may be applied into already stablished models [16,34,35], or in a new one to obtain the final information.

In the field of ethics, replacing the brain homogenate with the “animal free” (based on unfertilized chicken eggs) emulsion proposed in this work would speed up the procedures regarding the in vitro model proposed in 2013 by Mangas-Sanjuan et al. [17]. This acceleration is mainly due to the fact that, although using brain homogenate in the permeability tests reduces the number of animals used in research, an ethics committee approval is still needed, while, for using the egg emulsion, this approval is not necessary; i.e., for obtaining pig brain homogenate, it is necessary to sacrifice pigs, while for using eggs, no animal has to die.

Furthermore, in the economic and industrial field, the fulfillment of the 3Rs principles can also be beneficial because, generally, the time and the costs needed for applying the new alternative methods are much smaller than when using animals [36]. For instance, for applying the technique proposed in this paper, the industry needs the cells and the facilities for cell culture, which can be obtained after an initial investment, whereas, for applying the microdialysis, the maintenance of a stable and staff that take care of the animals is continuously necessary.

## 5. Conclusions

A new formulation (based on unfertilized chicken eggs) with the same proportion in proteins and lipids as a human brain has been developed in order to improve the ethics and reduce the costs of in vitro permeability tests. This formulation has proved to be a good alternative to brain homogenate in the preliminary study of drug distribution in the CNS, allowing researchers to obtain two different parameters (f_u,brain_ and V_u,brain_) in a quick and cheap way, as it is much more simple to gain access to eggs than to dead pig brains. Besides that, this methodology contributes to the protection of animals, as it replaces them successfully when, in an initial phase, the binding of a drug to the brain is studied. In this sense, the new formulation proposed here could be used in in vitro tests as a high throughput tool to select the most promising molecules and formulations in the early development of drugs for the treatment of CNS diseases; it is thus a great advance in the respectful use of animal lives.

## Figures and Tables

**Figure 1 animals-11-03521-f001:**
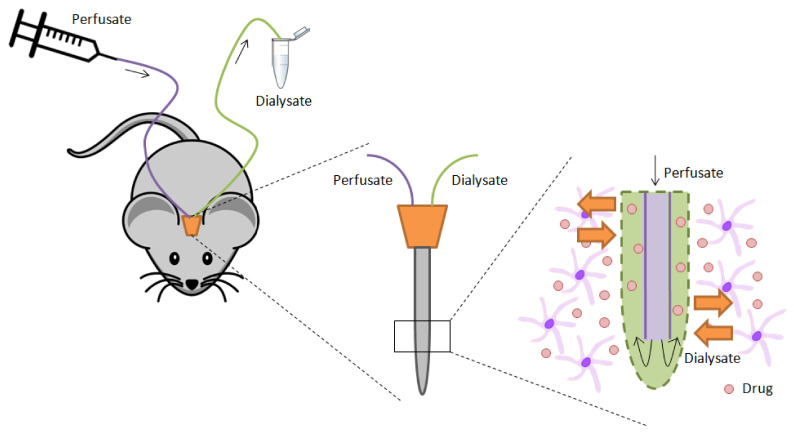
Scheme of brain microdialysis system.

**Figure 2 animals-11-03521-f002:**
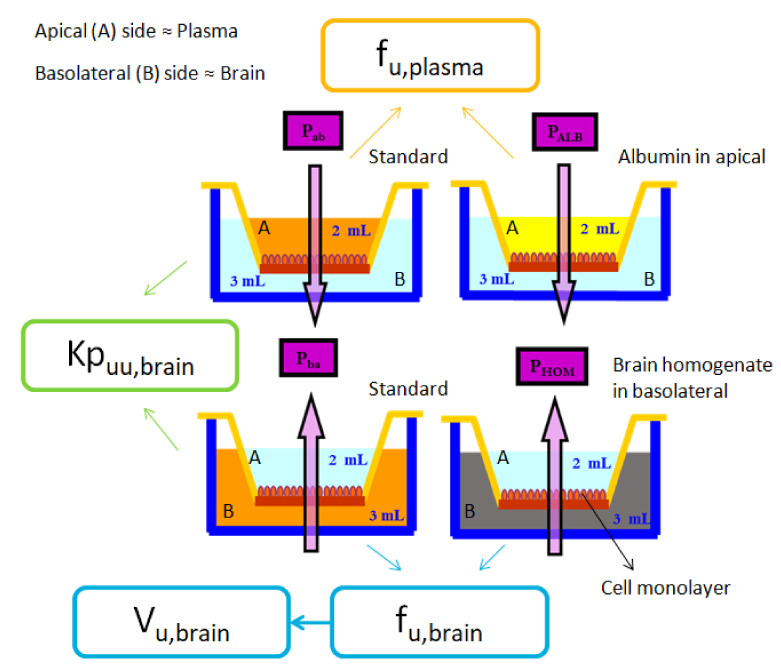
Scheme of the in vitro system with which the main parameters that describes the access and distribution of drugs in the CNS can be obtained. P_ab_: Apparent permeability from apical to basolateral in the standard experiment. P_ALB_: Apparent permeability from apical to basolateral in the experiment modified with albumin in apical. P_ba_: Apparent permeability from basolateral to apical in the standard experiment. P_HOM_: Apparent permeability from basolateral to apical in the experiment modified with brain homogenate.

**Figure 3 animals-11-03521-f003:**
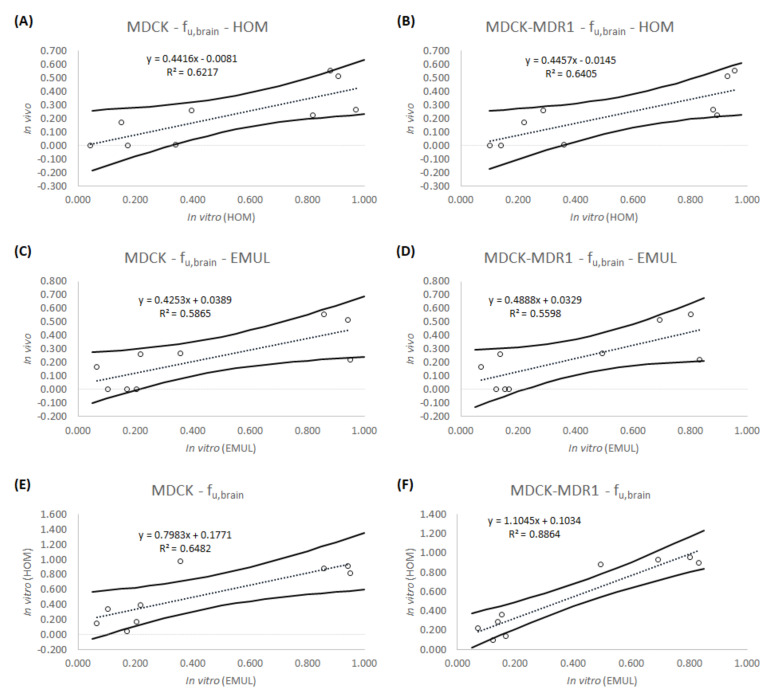
Correlations obtained for the f_u,brain_ parameter. (**A**) IVIVC between the in vitro parameter obtained using the brain homogenate and the MDCK cell line and the in vivo parameter. (**B**) IVIVC between the in vitro parameter obtained using the brain homogenate and the MDCK-MDR1 cell line and the in vivo parameter. (**C**) IVIVC between the in vitro parameter obtained using the new emulsion and the MDCK cell line and the in vivo parameter. (**D**) IVIVC between the in vitro parameter obtained using the new emulsion and the MDCK-MDR1 cell line and the in vivo parameter. (**E**) Relationship between the parameters predicted using the new emulsion and the parameters predicted using the brain homogenate in the MDCK cell line. (**F**) Relationship between the parameters predicted using the new emulsion and the parameters predicted using the brain homogenate in the MDCK-MDR1 cell line. Solid lines represent the 95% confidence interval.

**Figure 4 animals-11-03521-f004:**
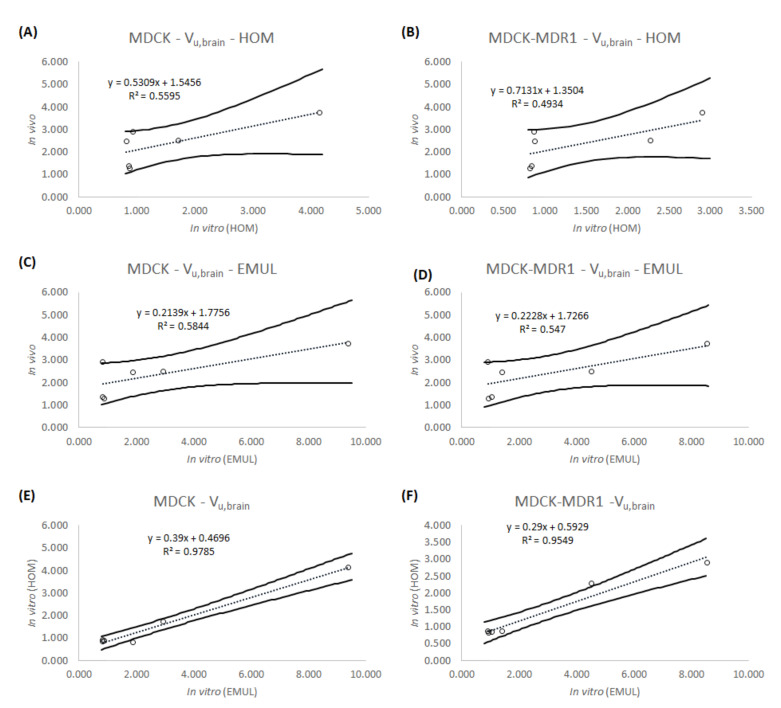
Correlations obtained for the V_u,brain_ parameter. (**A**) IVIVC between the in vitro parameter obtained using the brain homogenate and the MDCK cell line and the in vivo parameter. (**B**) IVIVC between the in vitro parameter obtained using the brain homogenate and the MDCK-MDR1 cell line and the in vivo parameter. (**C**) IVIVC between the in vitro parameter obtained using the new emulsion and the MDCK cell line and the in vivo parameter. (**D**) IVIVC between the in vitro parameter obtained using the new emulsion and the MDCK-MDR1 cell line and the in vivo parameter. (**E**) Relationship between the parameters predicted using the new emulsion and the parameters predicted using the brain homogenate in the MDCK cell line. (**F**) Relationship between the parameters predicted using the new emulsion and the parameters predicted using the brain homogenate in the MDCK-MDR1 cell line. Solid lines represent the 95% confidence interval.

**Table 1 animals-11-03521-t001:** Global prevalence of neurological disorders for people of all ages and people over 60 years in 2000 and 2019 [3].

Disease	Prevalence (Millions of People)							
	All Ages				60 to 89 Years			
	2000	2019	∆ (%)	Norm_∆ (%)	2000	2019	∆ (%)	Norm_∆ (%)
Alzheimer’s disease and other dementias	26.70	51.62	93%	54%	22.06	41.35	87%	13%
Parkinson’s disease	4.82	8.51	76%	41%	3.91	6.87	76%	6%
Other neurological disorders	0.04	0.06	45%	16%	0.01	0.02	92%	16%
Motor neuron disease	0.19	0.27	45%	15%	0.05	0.09	81%	10%
Multiple sclerosis	1.24	1.76	41%	13%	0.29	0.49	66%	1%
Schizophrenia	17.31	23.60	36%	9%	1.82	3.12	72%	4%
Idiopathic epilepsy	18.53	25.11	35%	8%	2.48	4.68	89%	14%
Migraine	852.24	1128.09	32%	5%	64.62	111.20	72%	4%
Tension-type headache	1524.6	1995.2	31%	4%	176.3	291.7	65%	0%
Mental disorders	777.26	970.07	25%	−1%	84.54	140.19	66%	0%
Neurological disorders	2016.6	2659.0	32%	5%	228.1	385.5	69%	2%

Mental disorders: schizophrenia, depressive disorders (major depressive disorder or dysthymia), bipolar disorder, anxiety disorders, eating disorders (anorexia nervosa, bulimia nervosa), autism spectrum disorders, attention-deficit/hyperactivity disorder, conduct disorder, idiopathic developmental intellectual disability and other mental disorders. ∆ expresses the increment in the prevalence of the disease from 2000 to 2019. Norm_∆ expresses the increment in the prevalence of the disease from 2000 to 2019 when the amount of people with that pathology in 2000 and 2019 is normalized by total amount of people in the world (from all ages and from 60 to 89 years old).

**Table 2 animals-11-03521-t002:** Molecular properties of the nine drugs tested [19,20].

Drug	MW(g/mol)	Solubility logS(pH 7)	logP	StrongestAcidic pKa	StrongestBasic pKa	Charge(pH 7.4)	Transporters(Substrates)
Amitriptyline	277.411	−1.63	4.81		9.76	+	ABCB1 (Pgp)
Atenolol	266.341	0.43	0.43	14.08	9.67	+	ABCB11
Carbamazepine	236.274	−3.79	2.77	15.96		0	ABCC2 RALBP1
Fleroxacin	369.344	−1.33	0.98	5.32	5.99	-	
Loperamide	477.050	−2.23	4.77	13.96	9.41	+	ABCB1 (Pgp)
Norfloxacin	319.336	−2.06	−0.97	5.58	8.77	0	ABCB1 (Pgp)
Pefloxacin	333.363	−1.21	0.75	5.5	6.44	-	ABCB1 (Pgp)
Propranolol	259.349	−1.03	2.58	14.09	9.67	+	ABCB1 (Pgp)
Zolpidem	307.397	−4.27	3.02		5.39	0	

MW = molecular weight.

**Table 3 animals-11-03521-t003:** Chromatographic conditions.

Drug	C (μM)	Wavelength	Mobile Phase	Retention Time (Min)
Amitriptyline	250	240 nm	40% Acid water 60% Acetonitrile	1.020
Atenolol	150	231 nm	20% Methanol 60% Acid water 20% Acetonitrile	1.330
Carbamazepine	150	280 nm	65% Acid water 35% Acetonitrile	1.926
Fleroxacin	150	285 nm	70% Acid water 30% Acetonitrile	1.348
Loperamide	241	260 nm	60% Methanol 40% Acid water	3.199
Norfloxacin	150	285 nm	70% Acid water 30% Acetonitrile	1.730
Pefloxacin	8.91	285 nm	65% Acid water 35% Acetonitrile	0.721
Propranolol	150	291 nm	30% Methanol 40% Acid water 30% Acetonitrile	1.950
Zolpidem	158	231 nm	60% Water 20% Methanol 20% Acetonitrile	4.624

Acid water had 0.05% (*v*/*v*) trifluoroacetic acid.

**Table 4 animals-11-03521-t004:** Apparent permeability obtained from the in vitro tests under different conditions (standard, brain homogenate or emulsion).

		MDCK Cell Line (×10^−6^ cm/s)			MDCK-MDR1 Cell Line (×10^−6^ cm/s)		
Drug	C (μM)	P_app B→A_	P_app HOM_	P_app EMUL_	P_app B→A_	P_app HOM_	P_app EMUL_
Amitriptyline	250	13.51	2.35	2.75	15.97	1.63	1.98
Atenolol	150	168.67	66.78	36.86	271.49	78.20	37.62
Carbamazepine	150	476.65	72.40	31.15	408.31	90.63	29.34
Fleroxacin	150	49.92 *	43.91 *	42.88	47.07 *	44.94 *	37.80
Loperamide	241	29.30	1.27	5.03	29.29	4.08	4.89
Norfloxacin	150	42.38	34.68	40.22	49.28	44.11	41.08
Pefloxacin	8.91	37.49 *	34.10 *	35.30	35.39 *	32.93 *	24.53
Propranolol	150	97.00	33.01	10.11	106.66	38.33	16.36
Zolpidem	158	36.48	35.42	13.03	33.43	29.46	16.52

* Data already published in [16].

**Table 5 animals-11-03521-t005:** f_u,brain_ predicted with the different experiments and in vivo f_u,brain_ values obtained in rat by Kodaira et al. and Friden et al. [25,26].

Drug		Rat	MDCK		MDCK-MDR1	
	C (μM)	f_u,brain_	f_u,brain HOM_	f_u,brain EMUL_	f_u,brain HOM_	f_u,brain EMUL_
Amitriptyline	250	0.002	0.174	0.204	0.102	0.124
Atenolol	150	0.261	0.396	0.219	0.288	0.139
Carbamazepine	150	0.170	0.152	0.065	0.222	0.072
Fleroxacin	150	0.555	0.880 *	0.859	0.955 *	0.803
Loperamide	241	0.002	0.043	0.172	0.139	0.167
Norfloxacin	150	0.222	0.818	0.949	0.895	0.834
Pefloxacin	8.91	0.514	0.910 *	0.942	0.931 *	0.693
Propranolol	150	0.005	0.340	0.104	0.359	0.153
Zolpidem	158	0.265	0.971	0.357	0.881	0.494

* Data already published in [16].

**Table 6 animals-11-03521-t006:** V_u,brain_ predicted with the different experiments and in vivo V_u,brain_ values obtained in rat by Kodaira et al. and Friden et al. [25,26].

Drug		Rat	MDCK		MDCK-MDR1	
	C (μM)	V_u,brain_	V_u,brain HOM_	V_u,brain EMUL_	V_u,brain HOM_	V_u,brain EMUL_
Atenolol	150	2.500	1.715	2.946	2.283	4.530
Carbamazepine	150	3.729	4.150	9.380	2.903	8.550
Fleroxacin	150	1.281	0.882	0.898	0.828	0.947
Norfloxacin	150	2.900	0.933	0.832	0.870	0.920
Pefloxacin	8.91	1.367	0.860	0.837	0.845	1.065
Zolpidem	158	2.464	0.818	1.880	0.881	1.414

## Data Availability

The authors confirm that the data supporting the findings of this study are available within the article and its Appendix A.

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
