# Peer review of "New In Vitro Methodology for Kinetics Distribution Prediction in the Brain. An Additional Step towards an Animal-Free Approach"

_animals, 2021, doi:10.3390/ani11123521_

Round 1
Reviewer 1 Report
This is an interest manuscript where the authors proposed a substitute model for in vitro PK study. They develop a new formulation to substitute the brain homogenate in the in vitro tests used for the prediction of drugs distribution in the brain. They found that the predictions are quite similar and when the new emulsion predictions and the brain homogenate predictions are represented together. I have some questions and suggestions regarding the content of the manuscript.
- How is the in vivo parameter obtained? Please update the method in the method sections
- Spell out the full term for IVIVC at the first time of use
- How is the linear regression conducted? What is the linear model used?
- Is the pig brain homogenate sterilized? How was is prepared, and what are the concentrations of drugs used? Please update in the method section
- According to the authors observation, the new formulation emulsion predicts quite similar results compared to the brain homogenate. However, many factors, other than the protein and lipid concentration may influence the drug distribution. Could the authors comment more on the pros and cons of the new model besides reducing animal use? Are there similar studies that use egg emulsion or other substitutes for such studies.
Author Response
Reviewer 1
This is an interest manuscript where the authors proposed a substitute model for in vitro PK study. They develop a new formulation to substitute the brain homogenate in the in vitro tests used for the prediction of drugs distribution in the brain. They found that the predictions are quite similar and when the new emulsion predictions and the brain homogenate predictions are represented together. I have some questions and suggestions regarding the content of the manuscript.
Thank you for your comments, we will try to address your questions.
- How is the in vivo parameter obtained? Please update the method in the method sections
The following information has been included in the method section:
Both in vitro parameters, fu,brain and Vu,brain were correlated with their correspondent value in vivo to obtain different in vitro-in vivo correlations (IVIVCs). The in vivo data were obtained from literature, specifically, from the following articles: [24,25]
- Spell out the full term for IVIVC at the first time of use
Done (line 246). Thank you.
- How is the linear regression conducted? What is the linear model used?
Thank you, a new section explaining the regression has been included.
2.6. In vitro-in vivo correlations (IVIVCs): linear regression.
Both in vitro parameters, fu,brain and Vu,brain were related with their correspondent value in vivo to obtain different in vitro-in vivo correlations (IVIVCs). The in vivo data were obtained from literature, specifically, from the following articles: [24,25]. The IVIVCs were adjusted to a linear model with the following structure: y = a +bx.
We also tried a model with a = 0, but r2 parameter was worse.
- Is the pig brain homogenate sterilized? How was is prepared, and what are the concentrations of drugs used? Please update in the method section
The pig brain homogenate was not sterilized, as, although the cells are grown under sterile conditions, the permeability study is carried out in a non-sterile environment and the brain homogenate is added in that moment. The concentrations of drugs used can be found in table 3 and the method for preparing the brain homogenate has been updated as follows:
The brain homogenate was obtained after triturating the pig brains that were kindly supplied by a local slaughterhouse and mixing them with phosphate buffer (180 mM, pH 7.4) solution in a ratio 1:3 (brain:buffer).
That ratio 1:3 was selected after trying other ratios 1:1 and 1:2, but the homogenate was to dense to take samples.
- According to the authors observation, the new formulation emulsion predicts quite similar results compared to the brain homogenate. However, many factors, other than the protein and lipid concentration may influence the drug distribution. Could the authors comment more on the pros and cons of the new model besides reducing animal use? Are there similar studies that use egg emulsion or other substitutes for such studies.
Thanks, that’s true, but, as far as we are concerned there are not studies with other substitutes to evaluate drug distribution in the brain so that’s the big novelty of our manuscript. We talked in our manuscript about the homogenation problem and later on we have also added information about the cons of differences between chicken and human brain, not in the proportion, but in the specific composition in lipids and proteins.
It cannot be forgotten that, the final aim of this type of studies is to try to guess what would happen if the drug is administered to human beings. On this respect, the differences on the specific composition (not in proportion) in lipids and proteins between chickens and humans may difficult the translation of data. Nonetheless, the development of PBPK models has proved to be a helpful tool in this process [16]. So, the data of this paper may be applied into already stablished models [16,34,35] or in a new one to obtain the final information.
Some advantages of our model are that the use of pigs instead of rats allows the use of a single animal for several studies, but the access to eggs is much easier, they can be bought at any supermarket. Furthermore, the use of eggs reduces costs in terms of the stable and the staff to take care of the animals.
Reviewer 2 Report
In their review, Bermejo et al. have described a model to predict the biodistribution and the fate of drugs targeting brain. Authors explain their findings in very detailed fashion. Their experiential results conveyed the message in detail. All the data looks promising and I would recommend the publication of this manuscript in Animals in its current form.
Author Response
In their review, Bermejo et al. have described a model to predict the biodistribution and the fate of drugs targeting brain. Authors explain their findings in very detailed fashion. Their experiential results conveyed the message in detail. All the data looks promising and I would recommend the publication of this manuscript in Animals in its current form.
Thank you very much for your evaluation.

Reviewer 3 Report
Thank you for your research, it is a very thorough piece of work and I think that there are some areas that need further clarity to ensure that your work is clearly understood and can be applied more widely.
Lines 19 and 20 state “The gold standard tool for obtaining this information is the microdialysis technique, but, according to 3Rs principles it would be better having an “animal-free” alternative technique.” I have two points here, the first is that I think you should clarify that microdialysis in this context refers to the use of animals (if this is true) and the second is that this goes beyond 3Rs principles and actually, the use of human biology based in vitro methods are vital for better understanding of human diseases, so I think that, whilst the immediate impact of your study does fit the 3Rs, you are underestimating the importance of your work by associating it only with the 3Rs.
Table 1 is very interesting, I would reorder it so that these are in descending order of prevalence as I think that would have more impact to the reader.
It would be nice to have more explanation of the data in the title- for example, what does the category “mental disorders” refer to specifically? Also, it seems that “other neurological disorders” has had the more dramatic increase with age since 2000 - what does this refer to?
I also think that this table needs to be put into context with the increase in the ageing population- it is not as simple as the increase in numbers of people with the condition, if there are more people of that age group in 2019 compared to 2000, then there is no effective increase in prevalence. Maybe you could express the date as a percent of the total population of that age group to try and normalise it and allow a clearer comparison?
Lines 62-70. I really like the explanation of microdialysis - it is quite a complex and technical procedure and the simple diagram is very clear and helpful.
Line 86. You mention the ethical issues with using animals for these tests and I agree with that but I think (as mentioned previously) that there are also issues with translation and the development of better in vitro methods are needed to improve this as well, perhaps you could add a comment along those lines here?
Lines 92-95. Your explanation of OLAW and ECVAM are slightly limited and I don’t think they add anything here, I am not sure what point you are using these to make. They have very different purposes!
Lines 108-113. I like the example you give of morphine and loperamide. The phrase “morphine crosses less the BBB and however, morphine is a much more potent drug” does not make sense to me and I think that this needs to be rephrased so your point is clear (I assume you are saying that Clin alone is not a good metric for potency?)
Figure 2 is quite complicated and I think it would benefit from a much more detailed legend to explain all the elements. Also – is this reproduced from another article with permission or is this your original figure (that is not clear to me and should be stated)
It would be interesting to know if the experiment in figure 2 had ever been carried out using human brain homogenate, either to compare to the animal data or as a stand alone, considering that we are usually more interested in addressing human disease. Are you aware if this has happened and could you comment (perhaps in the discussion?)
Line 138 – using chicken eggs is not the same as animal free, you need to clarify this!
Line 147 – can you confirm that these are chicken eggs please?
Line 184 – I am interested in the use of an orbital shaker, could you explain why this is required, please.
Could you explain the rationale for using the two cell lines, please? I am not familiar with MDCK-MDR and would appreciate some idea of what you expected with this version. You mention this in line 271-272 but I think it should be moved , perhaps to the introduction or methods. Thanks!
Supplementary data. Are there any canine data or other mammals that you could include? I am not sure of how much we’d expect data from a rat to mimic MDCK and so would be interested to know of other species-specific data if that is available. Of course, this is comparing in vitro and in vivo and different species, but more data would help improve confidence that your method is a valid way to move forwards. What if you used a rat brain cell line as the barrier in place of MDCK? Could that more closely mimic the in vivo rat data? Are there any existing studies that you could refer to here?
Lines 214-218 It is a bit confusing to the reader to begin the results with supplementary data that are not immediately accessible- is there no way that these data could be incorporated into the paper? Or maybe you could at least summarise these data rather than just referring to the tables, so the reader has an idea of the trends without having to leave the manuscript and find the supplementary data?
Did you measure/analyse the lipid and protein composition of the eggs used here (and the brain homogenate) or do you assume that these will be as you state in lines 268-269 in the Discussion? It’s not clear to me if these are assumed or actual measurements.
Your point in line 294 is very interesting about a lack of organised structure, and I wonder if you could comment further on the possible disadvantage(s) of using homogenates rather than intact tissues and how we might address this for the future.
Line 314 I am not sure what you mean by an animal-free slice? Do you mean using a brain slice? This could reduce animal use and/or improve welfare but it is not animal free! I feel that this needs more exploration as to how it complements this paper using a different homogenate and what information you would get from using a brain slice as I think these would be quite challenging experiments.
Line 321/330/345 – I reiterate that chicken eggs are not animal free!
Could the authors comment on any species- specific differences between chickens and humans that might impact on their data interpretation? If the ultimate species of interest is human, what do we need to consider in trying to translate these data to humans?
Lines 337-342. You make a good point about the needs for these approaches though and I agree with you regarding costs.
Line 346 – I think that could still be ethical arguments against what you are doing (chicken eggs could be argued to be animals, although of course they are not covered under the Directive 2010/63/EU) but I think that your work goes beyond ethics and the important point for me is the ease of access (much more simple to get eggs than dead pig brains!), costs implications and the similarity to human brain composition
Author Response
Thank you for your research, it is a very thorough piece of work and I think that there are some areas that need further clarity to ensure that your work is clearly understood and can be applied more widely.
Thank you very much for your extensive review work. We appreciate your comments and we will try to do our best to improve the manuscript.
- Lines 19 and 20 state “The gold standard tool for obtaining this information is the microdialysis technique, but, according to 3Rs principles it would be better having an “animal-free” alternative technique.” I have two points here, the first is that I think you should clarify that microdialysis in this context refers to the use of animals (if this is true) and the second is that this goes beyond 3Rs principles and actually, the use of human biology based in vitro methods are vital for better understanding of human diseases, so I think that, whilst the immediate impact of your study does fit the 3Rs, you are underestimating the importance of your work by associating it only with the 3Rs.
Thanks, we have tried to include in the abstract these important points, but we have limited space in that section. We have further commented the other aspects you mention later on in the introduction and conclusion sections.
- Table 1is very interesting, I would reorder it so that these are in descending order of prevalence as I think that would have more impact to the reader.
It would be nice to have more explanation of the data in the title- for example, what does the category “mental disorders” refer to specifically? Also, it seems that “other neurological disorders” has had the more dramatic increase with age since 2000 - what does this refer to?
I also think that this table needs to be put into context with the increase in the ageing population- it is not as simple as the increase in numbers of people with the condition, if there are more people of that age group in 2019 compared to 2000, then there is no effective increase in prevalence. Maybe you could express the date as a percent of the total population of that age group to try and normalise it and allow a clearer comparison?
Thank you very much for your comment. It has been really valuable. We have completed the information in table 1 and reordered the rows. We have extracted new conclusions from the table that will be very useful for the reader.
|
Prevalence (millions of people) |
|||||||||
|
All ages |
60 to 89 years |
||||||||
|
2000 |
2019 |
∆ (%) |
Norm_∆ (%) |
2000 |
2019 |
∆ (%) |
Norm_∆ (%) |
||
|
Alzheimer's disease and other dementias |
26.70 |
51.62 |
93% |
54% |
22.06 |
41.35 |
87% |
13% |
|
|
Parkinson's disease |
4.82 |
8.51 |
76% |
41% |
3.91 |
6.87 |
76% |
6% |
|
|
Other neurological disorders |
0.04 |
0.06 |
45% |
16% |
0.01 |
0.02 |
92% |
16% |
|
|
|
|
||||||||
|
|
|
||||||||
|
|
|
||||||||
|
Motor neuron disease |
0.19 |
0.27 |
45% |
15% |
0.05 |
0.09 |
81% |
10% |
|
|
Multiple sclerosis |
1.24 |
1.76 |
41% |
13% |
0.29 |
0.49 |
66% |
1% |
|
|
Schizophrenia |
17.31 |
23.60 |
36% |
9% |
1.82 |
3.12 |
72% |
4% |
|
|
Idiopathic epilepsy |
18.53 |
25.11 |
35% |
8% |
2.48 |
4.68 |
89% |
14% |
|
|
Migraine |
852.24 |
1128.09 |
32% |
5% |
64.62 |
111.20 |
72% |
4% |
|
|
Tension-type headache |
1524.6 |
1995.2 |
31% |
4% |
176.3 |
291.7 |
65% |
0% |
|
|
Mental disorders |
777.26 |
970.07 |
25% |
-1% |
84.54 |
140.19 |
66% |
0% |
|
|
|
|
||||||||
|
Neurological disorders |
2016.6 |
2659.0 |
32% |
|
228.1 |
385.5 |
69% |
|
|
Mental disorders: schizophrenia, depressive disorders (major depressive disorder or dysthymia), bipolar disorder, anxiety disorders, eating disorders (anorexia nervosa, bulimia nervosa), autism spectrum disorders, attention-deficit/hyperactivity disorder, conduct disorder, idiopathic developmental intellectual disability and other mental disorders.
∆ expresses the increment in the prevalence of the disease from 2000 to 2019.
Norm_∆ expresses the increment in the prevalence of the disease from 2000 to 2019 when the amount of people with that pathology in 2000 and 2019 is normalized by total amount of people in the world (from all ages and from 60 to 89 years old).
The GBD online results tool makes a group of “other neurological disorders” in which it includes disorders accounted for a low proportion of the overall burden, due to low prevalence or a less remarkable effect on life expectancy or functional abilities. The meaning of “mental disorders” has been included under the table.
We have also added some information about the data in the text:
According to table 1, in general, in the last decade, all the neurological disorders have increased their prevalence in at least 30%, but this increment gets much more pronounced in the population over 60 years old, with a minimum increment of 65% [3]. One could think that as the total population in the world has also increased with time, moving from 6143.5 million people in 2000 to 7713.5 million people in 2019 for all ages and from 602.7 to 996.7 million people for the group form to 60 to 89 years, the increments in prevalence may not be significant. Nonetheless, as it can be seen in the column Norm_∆ , where prevalences are normalized, there is a considerable increase in almost all the neurological disorders.
- Lines 62-70. I really like the explanation of microdialysis - it is quite a complex and technical procedure and the simple diagram is very clear and helpful.
Thanks, we appreciate a lot your positive comment.
- Line 86. You mention the ethical issues with using animals for these tests and I agree with that but I think (as mentioned previously) that there are also issues with translation and the development of better in vitro methods are needed to improve this as well, perhaps you could add a comment along those lines here?
We have added your point here:
Due to the high number of molecules and formulations to test, a great number of animals are required for these assays with the ethical problems that come with. Moreover, there are some issues with the translation of data from animals to human and, the use of human biology based in vitro methods is vital for better understanding of human diseases. So, the objective of the present work is to propose an innovative in vitro method amenable to high-throughput testing and which substitutes the use of brain animal/human homogenate in accordance to the 3 R’s principles (Replacement, Reduction and Refinement), which were established in the 20th century by Russell and Burch and which, nowadays, are strictly followed by scientists all over the world [9].
- Lines 92-95. Your explanation of OLAW and ECVAM are slightly limited and I don’t think they add anything here, I am not sure what point you are using these to make. They have very different purposes!
Thanks. We have removed that sentence.
- Lines 108-113. I like the example you give of morphine and loperamide. The phrase “morphine crosses less the BBB and however, morphine is a much more potent drug” does not make sense to me and I think that this needs to be rephrased so your point is clear (I assume you are saying that Clin alone is not a good metric for potency?)
Yes, that’s exactly what we wanted to express. We have changed the sentence.
Taking two opioid drugs, morphine and loperamide, as an example, and consulting the bibliography for its ability to access CNS, it can be seen that morphine (10.4 ± 3 μL/min × g brain) has a lower Clin than loperamide (98.6 ± 17.3 μL/min × g brain), so morphine has a lower permeability through the BBB [15,16]. Despite that, morphine is a much more potent drug, so, Clin alone is not a good metric for potency.
- Figure 2 is quite complicated and I think it would benefit from a much more detailed legend to explain all the elements. Also – is this reproduced from another article with permission or is this your original figure (that is not clear to me and should be stated)
It is an original figure. We have added the following legend under the figure caption. We do not specify the rest of parameters as they are defined in the paragraph above figure 2.
Pab: Apparent permeability from apical to basolateral in the standard experiment.
PALB: Apparent permeability from apical to basolateral in the experiment modified with albumin in apical.
Pba: Apparent permeability from basolateral to apical in the standard experiment.
PHOM: Apparent permeability from basolateral to apical in the experiment modified with brain homogenate.
- It would be interesting to know if the experiment in figure 2 had ever been carried out using human brain homogenate, either to compare to the animal data or as a stand alone, considering that we are usually more interested in addressing human disease. Are you aware if this has happened and could you comment (perhaps in the discussion?)
As far as we know, not human brain homogenate has been used in this kind of experiment.
- Line 138 – using chicken eggs is not the same as animal free, you need to clarify this!
Clarified.
Thereby, with the aim of improving the model and meeting the Replacement principle of the 3Rs, the purpose of this work was to develop a new formulation, based on unfertilized chicken eggs, to substitute the brain homogenate of the system and create an “animal-free” in vitro screening tool, able to predict both, the fu,brain and Vu,brain parameters.
- Line 147 – can you confirm that these are chicken eggs please?
Done
- Line 184 – I am interested in the use of an orbital shaker, could you explain why this is required, please.
We have further explained this point.
During the experiments, cells were maintained in an orbital shaker at 37 ºC and 100 rpm, so the agitation prevents the drug from precipitating and reduces the formation of a non-stirred layer over the cells, which would decrease the apparent permeability.
- Could you explain the rationale for using the two cell lines, please? I am not familiar with MDCK-MDR and would appreciate some idea of what you expected with this version. You mention this in line 271-272 but I think it should be moved , perhaps to the introduction or methods. Thanks!
The following information has been added to the method section.
The permeability studies were carried out in two different cell lines: MDCK and MDCK-MDR1. MDCK and MDCK-MDR1 cells come from the kidney of dogs, but when they are properly cultured, they form monolayers with quite strong tight junctions [22,30] and it is for that reason that they are accepted as appropriate tools to simulate the BBB, although they have not BBB transporters. In the MDCK-MDR1cell line, the issue of the lack of transporters is partially solved with the transfection with P-glycoprotein (Pgp), the most common efflux transporter in the BBB, so this line would be ideal for studying drugs with a passive access to the CNS, but also drugs which are substrate of Pgp, while MDCK would be better for studying passive drugs.
- Supplementary data. Are there any canine data or other mammals that you could include? I am not sure of how much we’d expect data from a rat to mimic MDCK and so would be interested to know of other species-specific data if that is available. Of course, this is comparing in vitro and in vivo and different species, but more data would help improve confidence that your method is a valid way to move forwards. What if you used a rat brain cell line as the barrier in place of MDCK? Could that more closely mimic the in vivo rat data? Are there any existing studies that you could refer to here?
As far as we know, the brain distribution studies tend to be done in rodents. Other cells lines, which simulate the BBB, from rat origin or human origin could be tested as previously done in
Sánchez-Dengra, B.; González-Álvarez, I.; Sousa, F.; Bermejo, M.; González-Álvarez, M.; Sarmento, B. In vitro model for predicting the access and distribution of drugs in the brain using hCMEC/D3 cells. Eur. J. Pharm. Biopharm. 2021, 163, 120–126, doi:10.1016/j.ejpb.2021.04.002.
Nonetheless, the use of, for instance, the hCMEC/D3 cell line makes the model more expensive and primary cell cultures which can be directly obtained from biopsies of other species are quite difficult to isolate without contamination (only 0.1% of brain are endothelial cells), and have a big variability intra and inter-laboratory. The rat is the cheapest and most used animal internationally. Not all the labs have installations for other animals and because of that there are a lot of correlations between in vitro tests and rats and between rats and humans.
- Lines 214-218 It is a bit confusing to the reader to begin the results with supplementary data that are not immediately accessible- is there no way that these data could be incorporated into the paper? Or maybe you could at least summarise these data rather than just referring to the tables, so the reader has an idea of the trends without having to leave the manuscript and find the supplementary data?
We have moved tables S1, S2 and S3 to the manuscript.
- Did you measure/analyse the lipid and protein composition of the eggs used here (and the brain homogenate) or do you assume that these will be as you state in lines 268-269 in the Discussion? It’s not clear to me if these are assumed or actual measurements.
We did not measure, we calculated the concentrations according to the information we found in the literature:
In an adult human, the CNS weight is around the 3% of the total human body weight [28] and, in terms of biochemical composition, the whole human brain has approximately 77-78% of water, 10-12% of lipids, 8% of proteins, 1% of carbohydrates, 2% of soluble organic substances and 1% of inorganic salts [29]. On the other hand, in a fresh egg, there are 12.5% of proteins (38% of them in the yolk and 62% in the whites) and 11.1% of lipids (all of them in the yolk) [21,30]. Taking into account these concentrations, a new formulation has been prepared in order to obtain an emulsion with the same composition as a human brain and the final concentration of protein has been 8% (5.1% from the yolk + 2.9% from the whites) and the concentration of lipids 12% (from the yolk).
- Your point in line 294 is very interesting about a lack of organised structure, and I wonder if you could comment further on the possible disadvantage(s) of using homogenates rather than intact tissues and how we might address this for the future.
We answer this question with question 17.
- Line 314 I am not sure what you mean by an animal-free slice? Do you mean using a brain slice? This could reduce animal use and/or improve welfare but it is not animal free! I feel that this needs more exploration as to how it complements this paper using a different homogenate and what information you would get from using a brain slice as I think these would be quite challenging experiments.
These two questions are related. Thanks. We wanted to say that, in the future we could move to develop another “animal-free” formulation to simulate brain slices, instead of brain homogenate. If we do so, we could compare the predictions about brain distribuition form a model in which the organised structure of the CNS would be kept intact with a more physiological substitute. We have completed, the paragraph about the slices to better explain ourselves.
Despite his success in the prediction of drug brain distribution, this type of in vitro model will not substitute the brain microdialysis technique, at least at the moment, because it cannot reflect all the physiological properties of an alive CNS. Maybe a future approach may be exploring the possibility of developing an organized animal-free slice with an organized structure which could be able to predict fu,brain the Vu,brain in a better way. To do this, first, the model should be tested substituting the brain homogenate by brain slices and evaluate its ability to predict brain distribution and then, if it is able to do it properly, the brain slices could be compared with the new animal-free slices. Nevertheless, as it is now, it is a useful tool that can be used in a complementary way when a new drug or an innovative delivery strategy is being developed. Thus, this model can be used as a rapid screening tool and its information, on its own, or combined with other one obtained from in silico [31–33] or PBPK [16,34,35] models could be used to move only a few selected candidates to in vivo studies.
- Line 321/330/345 – I reiterate that chicken eggs are not animal free!
We call it “animal-free” because not animals need to die for preparing the emulsion, but we have clarified that it based on unfertilized chicken eggs.
- Could the authors comment on any species- specific differences between chickens and humans that might impact on their data interpretation? If the ultimate species of interest is human, what do we need to consider in trying to translate these data to humans?
Done
It cannot be forgotten that, the final aim of this type of studies is to try to guess what would happen if the drug is administered to human beings. On this respect, the differences on the specific composition (not in proportion) in lipids and proteins between chickens and humans may difficult the translation of data. Nonetheless, the development of PBPK models has proved to be a helpful tool in this process [16]. So, the data of this paper may be applied into already stablished models [16,34,35] or in a new one to obtain the final information.
- Lines 337-342. You make a good point about the needs for these approaches though and I agree with you regarding costs.
Thanks
- Line 346 – I think that could still be ethical arguments against what you are doing (chicken eggs could be argued to be animals, although of course they are not covered under the Directive 2010/63/EU) but I think that your work goes beyond ethics and the important point for me is the ease of access (much more simple to get eggs than dead pig brains!), costs implications and the similarity to human brain composition
We have completed the conclusion to include those points.
A new formulation (based on unfertilized chicken eggs) with the same proportion in proteins and lipids as a human brain has been developed in order to improve the ethics and reduce the costs of in vitro permeability tests. This formulation has proved to be a good alternative to brain homogenate in the preliminary study of drugs distribution in the CNS, allowing researchers to obtain two different parameters (fu,brain and Vu,brain) in a quick and cheap way, as it is much more simple to access to eggs than to dead pig brains. Besides that, this methodology contributes to the protection of animals as it replaces them successfully when, in an initial phase, the binding of a drug to the brain wants to be studied. In this sense, the new formulation here proposed could be used in in vitro test as a high throughput tool to select the most promising molecules and formulations in the early development of drugs for the treatment of CNS diseases, being a great advance in the respectful use of animal lives.

Round 2
Reviewer 1 Report
The authors have addressed my points properly. I have no further comments.